# Over-Expressed GATA-1_S_, the Short Isoform of the Hematopoietic Transcriptional Factor GATA-1, Inhibits Ferroptosis in K562 Myeloid Leukemia Cells by Preventing Lipid Peroxidation

**DOI:** 10.3390/antiox12030537

**Published:** 2023-02-21

**Authors:** Silvia Trombetti, Nunzia Iaccarino, Patrizia Riccio, Raffaele Sessa, Rosa Catapano, Marcella Salvatore, Stelina Luka, Sergio de Nicola, Paola Izzo, Sante Roperto, Pasqualino Maddalena, Antonio Randazzo, Michela Grosso

**Affiliations:** 1Department of Molecular Medicine and Medical Biotechnology, University of Naples Federico II, 80131 Naples, Italy; 2Department of Veterinary Medicine and Animal Productions, University of Naples Federico II, 80137 Naples, Italy; 3Department of Pharmacy, University of Naples Federico II, 80131 Naples, Italy; 4Department of Physics, University of Naples Federico II, 80126 Naples, Italy; 5Centro Servizi Metrologici e Tecnologici Avanzati (CeSMA), University of Naples Federico II, 80126 Naples, Italy; 6Ceinge-Biotecnologie Avanzate Franco Salvatore, 80131 Naples, Italy

**Keywords:** ferroptosis, GATA-1 isoforms, glutathione peroxidase 4, myeloid leukemia, cell death, lipid peroxidation, RLS3

## Abstract

Ferroptosis is a recently recognized form of regulated cell death involving lipid peroxidation. Glutathione peroxidase 4 (GPX4) plays a central role in the regulation of ferroptosis through the suppression of lipid peroxidation generation. Connections have been reported between ferroptosis, lipid metabolism, cancer onset, and drug resistance. Recently, interest has grown in ferroptosis induction as a potential strategy to overcome drug resistance in hematological malignancies. GATA-1 is a key transcriptional factor controlling hematopoiesis-related gene expression. Two GATA-1 isoforms, the full-length protein (GATA-1_FL_) and a shorter isoform (GATA-1_S_), are described. A balanced GATA-1_FL_/GATA-1_S_ ratio helps to control hematopoiesis, with GATA-1_S_ overexpression being associated with hematological malignancies by promoting proliferation and survival pathways in hematopoietic precursors. Recently, optical techniques allowed us to highlight different lipid profiles associated with the expression of GATA-1 isoforms, thus raising the hypothesis that ferroptosis-regulated processes could be involved. Lipidomic and functional analysis were conducted to elucidate these mechanisms. Studies on lipid peroxidation production, cell viability, cell death, and gene expression were used to evaluate the impact of GPX4 inhibition. Here, we provide the first evidence that over-expressed GATA-1_S_ prevents K562 myeloid leukemia cells from lipid peroxidation-induced ferroptosis. Targeting ferroptosis is a promising strategy to overcome chemoresistance. Therefore, our results could provide novel potential therapeutic approaches and targets to overcome drug resistance in hematological malignancies.

## 1. Introduction

Ferroptosis is a recently described cell death process caused by the accumulation of lipid hydroperoxides (LPO) and inactivity of antioxidant systems such as glutathione peroxidase 4 (GPX4), a lipid repair enzyme containing selenocysteine (Sec) in their active site [1,2]. Polyunsaturated fatty acids-containing phospholipids (PUFA-PLs) are highly susceptible to lipid peroxidation under oxidative stress and can increase cell sensitivity to ferroptosis. Among long-chain PUFAs, arachidonic (20:4) (AA) and adrenic (22:4) (AdA) polyunsaturated omega-6 fatty acids in particular appear to be crucial for guiding cells into ferroptosis [3]. Several investigations have revealed connections between the interaction of lipid metabolism and ferroptosis with oncogenesis, tumor development, metastasis, and drug resistance. Growing evidence also indicates that ferroptosis induction may be a potential novel strategy for triggering cancer cell death, particularly for drug-resistant malignancies [4,5]. Interestingly, in the myeloid leukemia context, upregulation of GPX4 correlates with poor prognosis [6]. Therefore, targeting ferroptosis is emerging as a promising area of research in hematological malignancies [7,8,9]. GATA-1 is a key transcriptional regulator of hematopoiesis-related genes. A balanced expression of its two isoforms, the full-length GATA-1_FL_ and the shorter variant GATA-1_S_, contributes to controlling hematopoiesis, whereas their dysregulation alters the differentiation/proliferation potential of hematopoietic precursors with aberrant expression of GATA-1_S,_ representing an adverse prognostic factor in myeloid leukemia [10]. In our previous studies, we showed that GATA-1_S_ promotes cell survival and enhances apoptosis resistance in myeloid cells by modulating oxidative metabolism and cellular redox states [11,12]. Here, we provide the first evidence that overexpressed GATA-1_S_ inhibits ferroptosis in K562 myeloid leukemia cells by preventing lipid peroxidation. Lipidomic and functional analysis were conducted to elucidate these mechanisms. Our study contributes new understanding of the interplay between GATA-1_S_, lipid metabolism ^ and modulation of the redox state to inhibit cell sensitivity to ferroptosis in myeloid leukemia and could eventually provide novel potential therapeutic approaches and targets for hematological malignancies.

## 2. Materials and Methods

### 2.1. Cell Culture

The human K562 myeloid leukemia cell line provided by European Collection of Authenticated Cell Cultures (EACC #89122407). It was cultured in RPMI 1640 medium containing 10% fetal bovine serum (FBS) in addition to 4 mM glutamine, 10 U/mL penicillin, and 10 mg/mL streptomycin (Gibco, Thermo Fisher Scientific Inc., Waltham, MA, USA) at 37 °C in a humidified 5% CO_2_-containing atmosphere.

For transient transfection investigations, cells were grown at 60–70% confluency. Mycoplasma contamination in the cells was routinely examined using the PCR Mycoplasma Test Kit (AppliChem, Darmstadt, Germany #A3744,0020). Mycoplasma negative cells were utilized for experimental procedures.

### 2.2. Transient Transfection

As previously reported, transient transfection experiments were carried out in K562 cells using an Opti-MEM medium mixture (Invitrogen, Carlsbad, CA, USA) containing 1 μg of p3XFLAG-GATA-1_FL_ (GATA-1_FL_ cells), p3XFLAG-GATA-1_S_ (GATA-1_S_ cells), or p3XFLAG-CMV empty vector (mock control), and 5 μL of lipofectamine 2000 were used as a transfection reagent (Invitrogen) [11,13] Briefly, two hours before the transfection, cells were plated onto 6-well plates with 2 mL of serum-free medium at a density of 5 × 10^5^. After five hours transfection, FBS was supplemented to each well at a final concentration of 10%. Transfected cells were harvested for total RNA, protein extraction, and subsequent functional assays after 48 h.

### 2.3. Infrared Spectroscopy Analysis

IR analysis was performed by a Perkin-Elmer Spectrum 2000 FT-IR spectrometer (PerkinElmer Inc., Waltham, MA, USA). The cells samples were prepared by drop casting the solution on a p-type 500 µm thick silicon wafer. IR spectra were collected in the range 7800–400 cm^−1^ and averaged over 24 scans. The background spectrum was represented by the absorbance spectrum of the silicon substrate alone.

### 2.4. Fatty Acids Composition Analysis by Gas Chromatography–Mass Spectrometry

Total lipids were extracted from transfected cells by employing a lipid extraction kit (chloroform free) STA-612 (Cell Biolabs Inc., San Diego, CA, USA) according to the manufacturer’s instructions. Determination of the fatty acid profile was performed by analyzing the fatty acid methyl esters (FAMEs) obtained after transesterification as reported in literature [14]. Thus, the dried lipid extract was resuspended in 1000 μL of hexane and vortexed. Then, 2 aliquots of 500 μL were processed separately to generate two technical replicates. A total of 150 μL of methanolic KOH solution (2 N) were added to each aliquot and vigorously vortexed for 30 s. After shaking with a vortex, phase separation occurred. Then, 1 μL of the upper layer, containing the FAMEs, was injected in splitless mode into a GC-MS consisting of an Agilent 7890B GC (Agilent Technologies, Santa Clara, CA, USA) equipped with a Rxi-5MS 5% Phenyl 95% Dimethylpolysiloxane (29.7 m × 0.25 mm i.d., 0.25 um film thickness) capillary column (Restek, Bellefonte, PA, USA) with a time-of-flight mass spectrometer (LECO Corporation, Saint Joseph, MI, USA). The septum purge flow and purge flow were set to 3 and 20 mL/min, respectively. The injection port temperature was set to 280 °C. The initial temperature of the GC oven was set to 50 °C and held for 1.5 min, followed by heating at 20 min^−1^ until 80 °C and kept for 1 min, then increased to 300 °C at 10 min^−1^, making the total run time 36 min. Mass spectra were recorded in the range of 40–400 *m*/*z* with an acquisition rate of 10 spectra/s, and MS detector and ion source were switched off during the first 5 min of solvent delay time. The transfer line and ion source temperature were set to 280 and 250 °C, respectively. Helium (grade 6.0) was utilized as carrier gas, at a constant flow rate of 1.4 mL/min. The mass spectrometer was tuned in accordance with the manufacturer’s instructions using perfluorotributylamine (PFTBA). The raw GC- TOF-MS data was processed by the ChromaTOF software version 5.03.09.0 (LECO Corporation, Saint Joseph, MI, USA) that deconvolutes mass spectra and performs peak identification using the NIST11 library (NIST, Gaithersburg, MD, USA). Sample FAMEs peaks were first identified according to their similarity and retention index. The retention index of a chemical compound is its retention time normalized to the retention times of adjacently eluting n-alkanes (C8-C40). This index is independent of individual chromatographic systems (column length, film thickness etc.), allowing values measured by different laboratories to be compared. In fact, the calculated retention index is then compared with the one reported in NIST, and the ΔRI (which generally must not be higher than ±50) is generated (Appendix A). FAMEs assignment was further validated by employing a Supelco 37 Component FAME mixture (Supelco, Sigma-Aldrich, St. Louis, MO, USA).

### 2.5. Multivariate Data Analysis of FAMEs

Principal component analysis (PCA) [15] was employed for data exploration. ANOVA-simultaneous component analysis (ASCA) [16] was used to partition the variation present in the GC-MS data according to the study design factors, as previously reported by the authors [17]. The first factor corresponds to the different transfection vectors and contains three levels (GATA-1 _FL_, GATA-1_S_, mock vector); the second vector corresponds to the biological replicates (also defined as batch) and contains two levels (transfection experiment performed twice); their two-factor interaction effect (X_transfection_, X_batch_, X_transf×batch_ and X_residuals_) is then reported. The effect data matrices, X_transfection_ and X_batch_, were obtained from ASCA modelling; they were then autoscaled and modelled using PCA. The PCA and ASCA analyses were carried out using the PLS toolbox version 8.9 (Eigenvector Research, Manson, WA, USA) under MATLAB environment, version R2018b (MathWorks Inc., Natick, MA, USA) scripts.

### 2.6. Protein Extraction

K562 cells were harvested for total protein extraction and centrifuged at 3000× *g* for 10 min at 4 °C after two washes with 4 mL of cold 1× PBS. A lysis buffer (50 μL) containing 10% glycerol, 50 mM Tris-HCl pH 8.0, 150 mM NaCl, 0.1% NP-40, 1 mM EDTA pH 8, and 0.5 μL of protein inhibitor cocktail mixture (Sigma-Aldrich) were used to lyse the cells following incubation for 30 min on ice. After centrifugation at 10,000× *g* for 30 min at 4 °C, surnatants were collected in Eppendorf tubes. According to the Bradford method, protein concentration was evaluated spectrophotometrically using the Bio-Rad protein assay reagent (Bio-Rad Laboratories, Hercules, CA, USA).

### 2.7. Western Blot Analysis

Western blot analysis was carried out using 15 μg of whole cell protein extracts as previously described [11]. Proteins were separated in 10 or 12% SDS-page gels and electroblotted on nitrocellulose membranes by a Trans-blot Turbo instrument (Bio-Rad Laboratories). Following blotting, filters were treated with a blocking buffer (1X TBS, 0.1% Tween-20 with 5% *w/v* nonfat dry milk). Membranes were then hybridized either with the anti-FLAG antibody for 1 h and 30 min at 4 °C, or with the other primary antibodies overnight. The following experimental conditions were used: glutathione peroxidase 4/GPX4 (565320) (1:1000 dilution, Novus Biologicals Bio-Techne, Minneapolis, MN, USA #H00006391-M01), GAPDH (1:1000 dilution; Cell Signaling Technology, Danvers, MA, USA #2118), actin (C-11) (1:1000 dilution; Santa Cruz Biotechnology, Santa Cruz, CA, USA #sc-1615). After washing three times with 1× TBS-Tween 20 buffer for 5 min, filters were incubated for 45 min with secondary antibodies conjugated to peroxidase (Bio-Rad Laboratories). The ECL Immobilon Western Chemiluminescent HRP-substrate system (Millipore, Darmstadt, Germany) was used to detect the immunoreactive blots by autoradiography or by ChemiDoc XRS Image System (Bio-Rad Laboratories) in accordance with the manufacturer’s instructions. Actin and GAPDH were used as loading controls. The western blot bands were quantified using the ImageJ software.

### 2.8. Total RNA Extraction

As previously described, total RNAs were extracted from transfected K562 cells using the QIAzol reagent (Qiagen GmbH, Hilden, Germany) [11]. RNA was quantified by spectrophotometer analysis, and DNA contamination was excluded using a MOPS-buffered 1.5% agarose gel containing formaldehyde as a denaturing agent. After gel electrophoresis in MOPS 1× buffer (20 mM MOPS pH 7.0, 8 mM Sodium Acetate, 1 mM EDTA pH 8.0), the integrity of RNA was confirmed by UV exposure using a ChemiDoc XRS Image System (Bio-Rad Laboratories), according with the manufacturer’s instructions.

### 2.9. Quantitative Real-Time PCR Analysis

Using the QuantiTect Reverse Transcription Kit (Qiagen), 500 ng of total RNA previously extracted from K562 cells was transcribed into cDNA following the experimental conditions. In order to remove any DNA contaminations, 2 μL of 7× *g* DNA Wipeout Buffer were added to RNA in a final volume of 14 μL. Following incubation at 42 °C for 2 min, the mixture was placed immediately on ice. According to the kit protocol, the mixture was supplemented with 1 μL of RT primer mix, 4 μL of 5× Quantiscript RT Buffer, and 1 μL of Quantiscript Reverse Transcriptase. cDNA synthesis was obtained by incubating the reactions at 42 °C for 3 min and at 95 °C for 3 min in a preheated T100 Thermal Cycler (Bio-Rad Laboratories). cDNA samples were subsequently used for real-time reverse transcriptase PCR (qRT-PCR) procedures on a CFX96 Real-Time System (Bio-Rad Laboratories). According to GenBank sequences, primers for the GPX4 transcript were designed for quantitative real-time PCR analysis. Actin mRNA was used as reference control. All primer sequences are listed In Table 1. Real-time PCR reactions were prepared in triplicate for each sample. Total reaction volume was 20 μL that contained 10 μL of 2× SsoAdvanced Universal SYBR Green Supermix (Bio-Rad Laboratories), 0.38 μL of a 20 μM primer mix, 2 μL of cDNA (1/10 volume of qRT-PCR product), and 7.62 μL of nuclease-free water. Reaction running conditions were as follows: an initial denaturation of 30 s at 98 °C, followed by 40 cycles of amplification (95 °C for 15 s, 60 °C for 30 s) and a melting curve that was obtained as previously reported [11]. Three serial dilutions (1:10; 1:100; 1:1000) of the reverse transcriptase products were used to create the calibration curve in order to evaluate the efficiency of the PCR reaction. Real-time PCR reactions were run on a CFX96 Real-Time System (Bio-Rad Laboratories), and CT values were obtained from automated threshold analysis. The CFX Manager 3.0 software (Bio-Rad Laboratories) was used to analyze the data according to the manufacturer’s instructions. The acceptance range of the threshold cycles (Cq) resulted from the housekeeping GAPDH gene was set between 20.0 and 30.0. Therefore, samples with a GAPDH Cq > 30 were considered to have a low quality starting RNA sample and were not included in the analysis.

### 2.10. Lipid Peroxidation Assay

To visualize the lipid peroxidation in live cells, image-iT Lipid Peroxidation kit (Cat# C10445; Invitrogen) was used. This procedure is based on the use of BODIPY 581/591 C11 reagent, a sensitive fluorescent reporter that shifts from red to green in the presence of lipid peroxidation. Treatment of 48 h transfected cells with 100 μM menadione (Sigma-Aldrich) for 2 h was used to induce oxidative stress. Cells treated or untreated with menadione were then washed twice in PBS, plated on µ-Slide 8 wells (Cat: 80826, Ibidi GmbH, Gräfelfing, Germany), and incubated with 10 µM Image-iT Lipid Peroxidation Sensor that was added into the complete growth medium for 30 min at 37 °C, according to the manufacturer’s instructions. Hoechst 33342 reagent was added during the remaining 10 min incubation to stain live cells. For ferroptosis induction, cells were treated with RSL3, a specific inhibitor of GPX4 activity and expression [18]. A total of 48 h after transfection, cells were plated on µ-Slide 8 wells and treated by adding 20 μM 1S,3R-RSL3 (GPX4 inhibitor) or 20 μM 1R,3R-RSL3 (inactive enantiomer used as negative control of GPX4 inhibition) (Tocris-Biotechne, Minneapolis, MN, USA) into the complete growth medium for 2 h for microscopy assay or 24 h for real-time PCR, western blot, MTT assay, and Flow cytometry at 37 °C. Cells were washed three times with phosphate buffered saline (PBS) and then incubated with 10 µM Image-iT Lipid Peroxidation Sensor, as described above. A Leica Thunder Imaging System (Leica Microsystems Wetzlar, Germany) equipped with a LEICA DFC9000 GTC camera, Lumencor fluorescence LED light source and 63 × oil immersion objective was used to acquire Z-slice images. Quantitative fluorescence ratio analysis of green signals at 527 nm (representing peroxidized lipids)/red signals at 590 nm (representing nonperoxidized lipids) were quantified by ImageJ software.

### 2.11. Cell Viability and Cell Death Assays

Cell viability was determined using the MTT assay. Briefly, after transient transfection, K562 cells were seeded into a 96-well plate at a concentration of 1.5 × 10^4^ cells/100 μL. A total of 24 h after transfection, cells were incubated with 1S,3S RSL3 or 1R,3S RSL3 for an additional 24 h period, and then 10 μL of MTT labeling reagent (Cell Proliferation Kit 1, Roche, Mannheim, Germany) was added to each well, according to the procedures recommended by the manufacturer. The amount of the soluble formazan product in each well was measured by photometric reading at 570/690 nm using a Synergy H1 Hybrid Multi-Mode Microplate Reader (BioTek, Winooski, VT, USA). The experiments were repeated in triplicate for each transfection. Cell death analysis was assessed by double-staining with annexin-V and propidium iodide (PI) using an Annexin V-FITC Apoptosis Detection Kit 1 (BD Biosciences, Franklin Lakes, NJ, USA) according to the manufacturer’s protocol, as previously reported [11]. A total of 48 h after transfection, cells were analyzed using an Accuri C6 flow cytometer (BD Biosciences) and BD ACCURI C-Flow software.

### 2.12. Statistical Analysis

All data were calculated using the mean ± standard deviation (SD) of at least three separate experiments that were carried out in triplicate. The software utilized for data analysis was GraphPad Prism 7 (GraphPad Software, Inc., San Diego, CA, USA). The one-way analysis of variance technique and Dunnett’s multiple comparison test were used to compare the outcomes of mock control and treated cells in order to identify statistical differences. Differences were statistically significant when *p* < 0.05 (*) (#), and highly significant when *p* < 0.0001 (**) (##), versus each respective mock control or untreated control group.

## 3. Results

### 3.1. Infrared Spectroscopy Analysis

We recorded infrared (IR) absorbance spectra obtained from K562 cells transiently transfected with GATA-1_FL_ and GATA-1_S_ p3xFlag expression vectors (Appendix A). The IR spectra of the GATA transfected cells were ratioed against the spectrum of the mock sample to obtain differential spectra (Figure 1a,b) in the wavenumbers range from 2000 cm^−1^ to 4000 cm^−1^ (corresponding to a 2.5–5 μm wavelength range). IR absorbance spectra data from GATA-1_FL_ cells were found to be higher by nearly an order of magnitude compared with those obtained from GATA-1_S_ cells in the considered spectral range. As depicted in Figure 1, we observed an increase in the absorbance of GATA-1_FL_ cells in the higher-wavenumber region (3500–2550 cm^−1^), which is correlated to stretching vibrations, whereas the absorbance of GATA-1_S_ cells shows a significant decrease up to wavenumber 4000 cm^−1^.

As extensively discussed by Bellisola and Sorio [19], IR spectral features in the 3500–2550 cm^−1^) region could be related to structural features of lipid compounds. Therefore, the absorbance spectra variations detected in this IR region prompted us to hypothesize that the expression of specific GATA-1 isoforms could be associated with differences in cell lipid profiles.

### 3.2. Fatty Acids Composition

Based on these observations, we sought to assess lipid profiles associated with specific GATA-1 isoforms. Determination of the fatty acid profile was performed by gas chromatography–mass spectrometry analysis on total lipids extracted from GATA-1_FL_ and GATA-1_S_ cells. The deconvolution of the GC-MS data revealed a total of 33 tentatively assigned FAMEs. However, we decided to rely solely on the peaks that were also confirmed by the FAMEs standards mixture. Thus, a total of 23 compounds (22 FAMEs and cholesterol) were included in the subsequent data analysis (Appendix A). The final dataset, consisting of 12 samples (three transfection vectors × two batches × two technical replicates) and 23 variables (compounds), was explored using Principal Component Analysis, which showed the presence of a great experimental variability, especially among the batches, as reported in Appendix A. Thus, to remove the unwanted variation stemming from the batches, a chemometric tool called ASCA was employed. This algorithm allowed us to filter out the data variation due to the batches and focus on the variation caused by the different GATA-1 isoforms by partitioning the original GC-MS data matrix into four new data matrices, each one containing the variation caused by the transfection vector, batch, transfection vector–batch interaction, and individual variability (residuals) (output details are reported in Appendix A). The ASCA results showed, as expected, that the batch had the biggest effect and explained 53.71% of total variance, implying that there is a significant variation in the biological replicates prepared at two different times. Nevertheless, the transfection vector effect explained 19.21%, and it was statistically significant. The significance of the *X_transf×batch_* interaction term suggests a possible difference of the transfection vector across the two batches. Since this interaction effect explained only a little variation, and since the purpose of this study was directed towards a difference among the various transfection vectors, this term was not further examined. The variation retained in the residual matrix E accounts for the sum of total uncertainty derived from both the individual differences between the samples and the experimental errors. The score plot obtained after this analysis showed a much clearer separation of the three groups of transfected cells (Appendix A) compared to the separation obtained by the PCA reported in Appendix A. However, this model mainly explained the difference among the mock vector group and cells transfected with GATA-1 expression vectors (GATA groups) and did not allow the differences between GATA-1_FL_ and GATA-1_S_ cells to be precisely identified. Thus, in the next step, the mock vector group was removed from the analysis, and ASCA was recomputed on the reduced data matrix (containing only data from GATA-1_FL_ and GATA-1_S_ cells). The output of this new analysis (details are reported in Appendix A) further confirmed the significance of the transfection vector effect, thus allowing for the extrapolation of the actual differences in the fatty acids composition of the cells as characterized by the two GATA-1 isoforms. Indeed, the obtained PC1 scores and loading plots (Figure 2) clearly showed a separation between the two investigated groups. In particular, cells transfected with GATA-1_FL_ vector resulted in a higher presence of saturated fatty acids (SFAs), such as C8:0, C9:0, C14:0, C15:0, C17:0, C20:0, and C25:0, as well as polyunsaturated fatty acids (PUFAs), such as C18:2, C20:4, C20:5, C22:6, C22:5, and C22:4, with all of them having positive loadings on PC1 (surrounded by a red square in the upper part of the loadings plot reported in Figure 2b) when compared to the GATA-1_S_ group, which was characterized by low levels of such compounds. Moreover, GATA-1_FL_ cells showed increased levels of some monounsaturated fatty acids (MFAs) such as C16:1, C17:1, and C22:1, as well as cholesterol. At the same time, the transfection with GATA-1_S_ vector was correlated with increased levels of three SFAs, namely C12:0, C16:0, and C18:0, as well as of C18:1 (both E and Z isomers), all of which had negative loadings on PC1 (surrounded by a green square in the lower part of the loadings plot reported in Figure 2b).

### 3.3. Correlation between GATA-1 Isoforms Expression and Lipid Peroxidation

The most significant substrate causing ferroptosis is produced when PUFA-PLs are oxidized to produce lipid hydrogen peroxide (PUFA-PL-OOH). According to this evidence, results from lipidomic analysis raised the possibility that GATA-1_FL_ and GATA-1_S_ cells could be susceptible to ferroptotic cell death in different ways, given the higher content of key drivers of ferroptosis, including AA (20:4) and AdA (22:4), in GATA-1_FL_ cells with respect to GATA-1_S_ cells. Therefore, since lipid peroxidation is a hallmark of ferroptosis, we began testing our hypothesis by staining cells with Image-iT Lipid Peroxidation Sensor, a probe that allows fluorescence changes from red to green when bound to peroxidized lipids. Results collected with a Leica Thunder Imaging System demonstrated that, even under basal conditions, GATA-1_FL_ cells showed a higher rate of lipid peroxidation with respect to mock as well as to GATA-1_S_ cells; such differences were even more evident under pro-oxidant conditions triggered by menadione (Figure 3). Conversely, fluorescent signals corresponding to the reduced form of the peroxidation sensor in GATA-1_S_ cells indicated a lower level of lipid peroxidation, even under pro-oxidant conditions, thus supporting our working hypothesis.

### 3.4. Correlation between GATA-1 Isoforms and Expression Levels of Glutathione Peroxidase 4 (GPX4)

GPX4 is the only enzyme that can reduce phospholipid peroxides (PL-OOH), which is done by using two reduced glutathione molecules to convert peroxides into corresponding inactive redox lipid alcohols (PL-OH), and thereby protecting cells from ferroptosis [22]. However, when GPX4 is downregulated, PL-OOHs levels may increase, resulting in lipid peroxidation damage to the membrane and ferroptosis-mediated cell death [18,23]. Accordingly, upregulation of GPX4 has been reported to function as a major suppressor of ferroptosis in cancer cells [24]. Therefore, given the central role played by the glutathione–GPX4 axis in limiting lipid peroxidation, we questioned whether GATA-1_S_ might promote ferroptosis resistance by enhancing these antioxidant defenses. It is noteworthy that we had previously found increased GSH levels in GATA-1_S_ cells, mainly in its reduced form, indicative of a stronger antioxidant capacity in these cells [11]. In the present study, we examined whether GATA-1 isoforms contribute differently to GPX4 expression. To this aim, we first analyzed GPX4 expression levels on protein extracts from K562 cells that were transiently transfected either with p3xFLAG expression vectors for specific GATA-1 isoforms (GATA-1_FL_ and GATA-1_S_ cells) or an empty p3xFLAG vector as mock control. After 48 h, transfected cells were collected and GPX4 protein levels were evaluated by western blot analysis on a 12% SDS-page gel. As shown in Figure 4a,b, overexpression of the GATA-1_S_ isoform was accompanied by markedly increased GPX4 protein level with respect to both GATA-1_FL_ and mock cells. To determine if the increased protein levels were related to an increase in GPX4 mRNA levels, a quantitative real-time PCR assay was conducted. As shown in Figure 4c, the qRT-PCR analysis revealed higher levels of the GPX4 transcript that may be due to transcriptional activation or to decreased mRNA degradation. Although the molecular mechanisms underlying GATA-1-mediated regulation of GPX4 expression remain to be clarified, these findings demonstrate that overexpression of GATA-1_S_ is associated with enhanced antiperoxidation defenses in K562 myeloid cells.

### 3.5. Effects of GPX4 Inhibition on Lipid Peroxidation in Cells Expressing GATA-1 Isoforms

Enzyme inhibitors or suppressors of GPX4 expression can be used to block GPX4 activity and promote ferroptosis. Among enzyme inhibitors, RSL3 can specifically block GPX4 action by covalently binding to its active site, thus acting as a ferroptosis-triggering agent by causing ROS accumulation, lipid peroxidation, and eventually, cell death [4,25,26,27,28]. Therefore, to further explore the possibility that GATA-1s can activate antiferroptotic mechanisms, we tested the effects on lipid peroxidation following GPX4 inhibition by RSL3 treatment in K562 cells with overexpressing GATA-1 isoforms.

Transfected cells with GATA-1 isoforms were treated for 24 h with 20 μM 1S,3R-RSL3 (thereafter RSL3), a specific GPX4 inhibitor, or with its inactive enantiomer 1R,3R-RSL3, used as a negative control. As expected, results showed that GPX4 inhibition renders cells more susceptible to lipid peroxidation. However, this effect was dramatically more enhanced in GATA-1_S_ cells after exposure to RSL3 compared to the inactive enantiomer treatment (Figure 5). These findings are in striking contrast with results obtained under pro-oxidant conditions (menadione treatment), in which no significant increase in peroxidized lipid content in GATA-1_S_ cells was observed compared to the GATA-1_FL_ counterpart (Figure 3). These findings indicate that, although GATA-1_S_ expression can lead to the development of protective mechanisms against oxidative stress, the loss of antioxidant defenses renders these cells more vulnerable to oxidative processes, eventually leading to an increased rate of ferroptotic cell death.

### 3.6. Effects of GPX4 Inhibition on Cell Viability and Ferroptotic-Mediated Cell Death in Cells Overexpressing GATA-1 Isoforms

Based on the effects of GPX4 inhibition on lipid peroxidation in K562 cells overexpressing GATA-1 isoforms and the proferroptotic effects reported to be related to GPX4 inhibition, we thus hypothesized that RSL3 treatment could have a more dramatic impact on reduced cell viability and increased rate of cell death in GATA-1_S_ cells with respect to cells overexpressing the full-length isoform of GATA-1. Therefore, with the aim of verifying this hypothesis, we next examined the effects of RSL3 treatment on cell viability and cell death rates in GATA-1_FL_ and GATA-1_S_ cells. Initially, an MTT assay was performed to monitor the status of cell viability after exposure to RSL3 treatment [29]. Results showed a significant marked reduction in cell viability in GATA-1_S_ following inhibition of GPX4 activity compared to untreated and mock controls. Conversely, no effects were observed when cells were treated with the inactive RSL3 enantiomer (1R,3R), thus confirming that these effects are strictly related to GPX4 inhibition (Figure 6).

GPX4 downregulation and lipid peroxidation are significant hallmarks of ferroptosis-related cell death. Ferroptosis is a form of regulated necrotic cell death that can be identified through propidium iodide (PI) staining [30,31,32,33,34]. Thus, according to this evidence, we performed flow cytometry analysis using annexin-V/PI staining and detected significantly increased cellular necrosis (% annexin-V+/PI+ and annexin-V-/PI+ cells) in cells treated with RSL3 compared with untreated and negative control cells. Notably, a more significant increase in the number of PI+ cells was observed in GATA-1_S_ compared with mock and GATA-1_FL_ cells (Figure 7). Anyway, although GPX4 downregulation and lipid peroxidation unequivocally support ferroptotic-regulated cell death processes, accumulating evidence indicates the existence of extensive cross-talks between different modes of cell death, including apoptosis. This can explain the relative enrichment in annexin-V+/PI- cells, which is even more evident in mock and GATA-1_FL_ cells (LR quadrants in Figure 7a). Collectively, these findings indicate that RSL3 makes GATA-1_S_ cells more sensitive to necrotic cell death pathways thus suggesting GPX4 inhibition as a possible target to overcome cell death resistance observed in cells overexpressing the GATA-1_S_ isoform [11].

Since the existing research indicates that exposure to RSL3 is accompanied by a decrease in the expression of GPX4 [28], with the aim to further analyze the mechanisms underlying the downregulated antioxidant response following RSL3 treatment in GATA-1_FL_ and GATA-1_S_ cells, we examined possible changes in GPX4 expression levels in these cell samples. As shown in Figure 8a–d, western blot analysis showed that RSL3 treatment was accompanied by a dramatic reduction of GPX4 protein levels in all treated samples (~80–95% reduction with respect to the corresponding untreated cells). Notably, this effect appeared to be particularly consistent in GATA-1_S_ cells, given their higher GPX4 steady-state levels. Next, a quantitative real-time PCR assay was used to evaluate the levels of the GPX4 transcript. As shown in Figure 8e, results showed reduced levels of the GPX4 transcript in all cell samples exposed to RSL3 treatment, with more consistent effects in GATA-1_S_ cells, in line with the western blot data. Together, these findings are thus in agreement with literature reporting that the inhibitory effects mediated by RSL3 comprise reduced activity along with lowered protein and transcriptional levels of GPX4 [28].

## 4. Discussion

Lipids are crucial elements of biological membranes and play a key role in energy storage and signal transmission within cells. Recently, membrane lipids have emerged as important regulators of ferroptosis, a nonapoptotic cell death process. Ferroptosis is triggered by the accumulation of oxidized polyunsaturated fatty acyl moieties in the context of phospholipid membranes. In this context, several lines of evidence indicate that altered lipid composition can affect ferroptosis sensitivity with high saturation levels of membrane phospholipids making cells less susceptible to ferroptosis [35]. In more recent years, ferroptosis has been shown to play an important role in many cancer types, including hematological neoplasms, such as leukemia [36]. In this context, although the role of aberrant lipid metabolism has been extensively investigated in solid tumors, it has not yet sufficiently elucidated in hematological malignancies. Nevertheless, since ferroptosis has been proved to eradicate tumor cells that are resistant to a variety of anticancer drugs, the use of ferroptosis as a target for the prevention and treatment of hematological malignancies has emerged as a challenging area of research. A number of ferroptosis-inducing compounds, such RSL3 and erastin, have been shown to be highly effective in killing cancer cells in vitro, however their pharmacokinetic characteristics, such as their solubility and metabolic stability, make them unsuitable for usage in vivo. Therefore, a better understanding of the ferroptosis regulatory network and physiological role is required in order to develop more effective cancer-specific strategies for inducing ferroptosis.

According to the evidence that the transition from a normal to a cancerous phenotype is associated with biochemical, molecular, and morphological changes, as thoroughly reviewed by Bellisola and Sorio in 2012, significant efforts have been focused on the implementation of optical techniques capable of determining dimension, shape, or other physical cell properties that can be used for early detection of neoplastic changes, and thereby improve survival and clinical outcomes in cancer patients. We recently made use of this approach to find novel oncometabolites and biomarkers connected to the expression of different isoforms of the transcriptional factor GATA-1 in the myeloid leukemia K562 cell line, namely GATA-1_FL_ (GATA-1_FL_ cells) or GATA-1_S_ (GATA-1_S_ cells), respectively, two GATA-1 isoforms that play opposite roles in the differentiation and proliferation processes of several hematopoietic lineages. In fact, whereas GATA-1_S_ is involved in the maintenance of the proliferative potency of hematopoietic precursors, GATA-1_FL_ promotes their terminal differentiation [37,38]. A prevalence of GATA-1_S_ has been found to be associated with hematological malignancies, including myeloid leukemia, where persistent high GATA-1_S_ levels represent a poor prognostic factor [39,40,41]. However, although several reports emphasize the proleukemic role of this isoform, mechanistic details still need clarification to decipher its contribution to malignant hematopoiesis.

Because light scattering may be more sensitive to morphological alterations than other currently utilized imaging techniques, optical techniques are used to detect biochemical and morphological markers that are associated with precancerous conditions [19]. Recently, this approach led us to demonstrate that GATA-1_FL_ and GATA-1_S_ isoforms specifically contribute to inhibiting or promoting proliferative pathways through different modulations of oxidative stress conditions [11]. Subsequently, we were also able to emphasize the effects of GATA-1_S_ on the generation of mitochondrial and cytoplasmatic ROS as well as on the modulation of oxidative phosphorylation through the control of the expression levels of SDHC (subunit C of the succinate dehydrogenase complex II of the respiratory chain). Given the role played by ROS in myeloid leukemogenesis, these observations led us to highlight the link between oxidative stress conditions triggered by GATA-1_S_, increased cell proliferation rate, and cell death resistance [12].

In the present study, spectral differences in the 3500–2550 cm^−1^ IR region were found in GATA-1_FL_ cells and GATA-1_S_ cells. According to literature data indicating the correspondence between vibrational signals in this IR region and structural features of lipid compounds [19], we thus speculated that differential expression of GATA-1 isoforms could be associated with specific lipid profiles. To start exploring this hypothesis, lipidomic analysis was performed in GATA-1_FL_ and GATA-1_S_ cells that revealed significant variations between the two investigated groups. Particularly, GATA-1_FL_ cells resulted in a higher presence of PUFAs, such as C18:2, C20:4, C20:5, C22:6, C22:5, and C22:4, compared to the GATA-1_S_ group which was, instead, characterized by low levels of such compounds. Given the reported role of PUFAs, particularly arachidonic (C20:4) and adrenic (C22:4) acids, in promoting ferroptotic sensitivity, we thus postulated that the lipid content associated with GATA-1_S_ overexpression could render cells less prone to ferroptosis-related membrane lipid peroxidation. In agreement with these observations, we were able to demonstrate that GATA-1_S_ cells exposed to oxidative stress showed reduced membrane lipid peroxidation, thus supporting our working hypothesis that GATA-1_S_ expression may correlate with reduced cellular sensitivity to ROS-mediated ferroptosis. Given the central role played by the glutathione-GPX4 axis in limiting lipid peroxidation, we also questioned whether GATA-1_S_ expression might further promote ferroptosis resistance by enhancing these antioxidant defenses. Gene expression studies in these cells revealed increased GPX4 levels in GATA-1_S_ cells with respect to GATA-1_FL_ cells. In this context, although there is no evidence that the GPX4 gene is a direct target of GATA-1 (GeneCards database, www.genecards.org), given the extensive GATA-1-regulated network of gene activation and repression, we speculate that GPX4 upregulation associated with GATA-1_S_ can be explained by a mechanism of GATA-1-mediated indirect transcriptional regulation. Interestingly, it should be noted that we had previously found increased GSH levels and enhanced expression of enzymes involved in GSH biosynthesis in cells overexpressing GATA-1_S_ [11,12]. Therefore, collectively, results provided in this paper further reinforce our previous findings which show that cells overexpressing GATA-1_S_ exhibit great antioxidant capacity as a mechanism to sustain cell survival.

Here, we add further light on the proleukemic role of GATA-1_S_ by providing the first evidence that its overexpression can be associated with reduced sensitivity to ferroptosis. We also evaluated the effects of GPX4 inhibition as a strategy to overcome cell death resistance shown by GATA-1_S_ cells. In these cells, treatment with RSL3, a specific GPX4 enzyme inhibitor causing ROS accumulation and lipid peroxidation, was accompanied by enhanced membrane lipid peroxidation and reduced GPX4 expression levels, thus increasing the rate of ferroptosis-mediated cell death.

Together, our results shed novel light on the molecular mechanisms by which GATA-1_S_ could contribute to leukemia onset and development by demonstrating that its dysregulated expression is associated with altered lipid metabolism, enhanced antioxidant activities, and reduced sensitivity to ferroptosis to sustain proliferation programs and survival pathways in normal or malignant hematopoiesis.

## 5. Conclusions

Induction of ferroptosis is a promising area for further study in hematological malignancies. Since drug resistance has always been one of the key elements influencing a patient’s prognosis, it is expected that targeting ferroptosis could overcome the resistance of traditional chemotherapy and targeted therapy in hematological malignancies. Thus, it is anticipated that focusing on ferroptosis would help treat hematological malignancies despite the resistance to targeted therapy and conventional chemotherapy.

Given the proleukemic role of GATA-1_S_, our findings provide the first evidence that overexpressed GATA-1_S_ prevents myeloid leukemia cells from lipid peroxidation-induced ferroptosis. Notably, although p53 has been identified as a relevant player in ferroptosis, mounting research results indicate p53-independent ferroptosis pathways in several experimental models. Therefore, our results obtained in p53-deficient K562 cells further support the role of alternative pathways that may trigger or inhibit ferroptosis and the relevance of GATA-1 in regulating these processes. A better understanding of the interplay between GATA-1_S_, lipid metabolism, and modulation of the redox state to inhibit cell sensitivity to ferroptosis in myeloid leukemia could be fundamental to provide novel potential therapeutic approaches and targets to overcome chemoresistance in hematological malignancies.

## Figures and Tables

**Figure 1 antioxidants-12-00537-f001:**
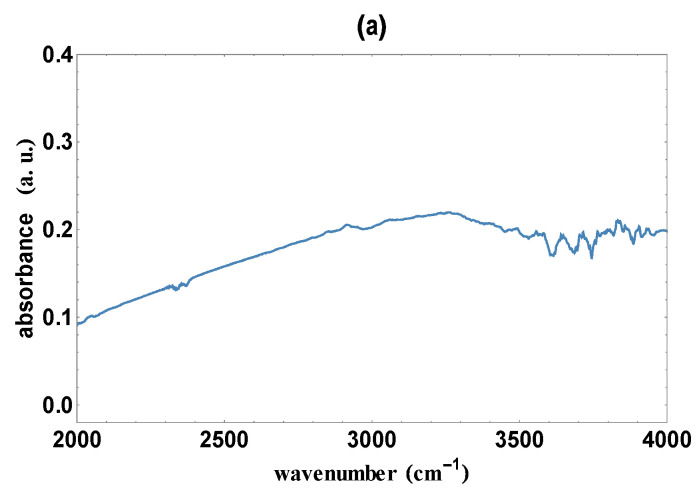
Infrared absorbance spectra of K562 cells transiently transfected with (**a**) GATA-1_FL_ and (**b**) GATA-1_S_ expression vectors. Spectra wavenumbers are represented on the abscissa and the absorbance is plotted on the ordinate scale.

**Figure 2 antioxidants-12-00537-f002:**
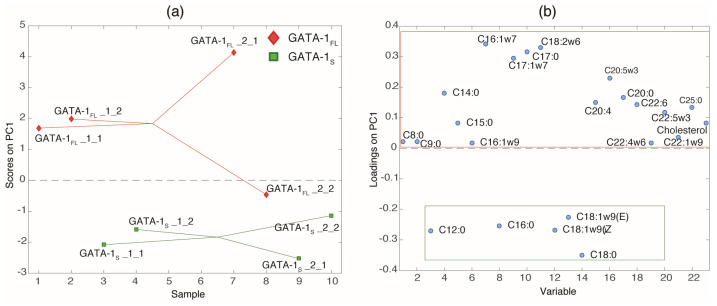
Comparative lipidomic analysis revealing significant changes in fatty acid content in GATA-1_FL_ and GATA-1_S_ groups. The figure shows (**a**) scores and (**b**) loadings plots derived from the ASCA output performed by including only GATA-1_FL_ and GATA-1_S_ groups.

**Figure 3 antioxidants-12-00537-f003:**
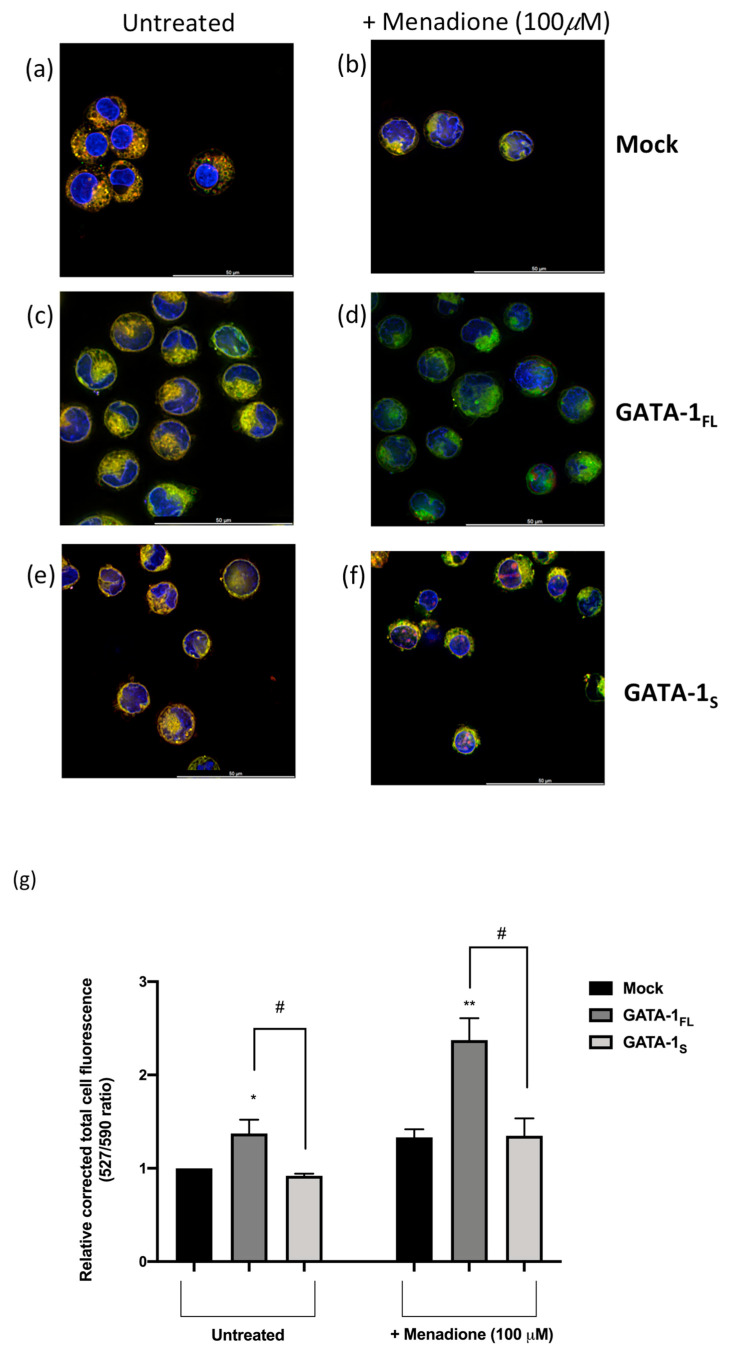
Lipid peroxidation in K562 cells transfected with GATA-1 isoforms. Cells were stained with BODIPY 581/591 C11 reagent for measurement of lipid peroxidation: the red to green shift corresponds to lipid peroxidation. Nuclei of live cells were stained with Hoechst 33342 (blue). (**a**–**f**) Representative fluorescence merged images of K562 cells stained with BODIPY™ 581/591 C11 reagent after exposure to RSL3 for 24 h to assess the generation of lipid peroxidation under basal (**a**,**c**,**e**) and pro-oxidant conditions induced by menadione treatment (**b**,**d**,**f**). (**g**) Quantitative fluorescence ratio analysis of green signals at 527 nm (representing peroxidized lipids)/red signals at 590 nm (representing nonperoxidized lipids) quantified by ImageJ, as elsewhere reported [20,21]. Scale bar = 50 μm. All data shown are representative of three independent experiments. Statistical analysis was performed by one-way ANOVA, followed by Dunnett’s multiple comparisons test, where appropriate. * *p*-value ≤ 0.05, and ** *p*-value ≤ 0.001 compared to untreated control group, # *p*-value ≤ 0.05, GATA-1_FL_ cells versus GATA-1_S_ cells.

**Figure 4 antioxidants-12-00537-f004:**
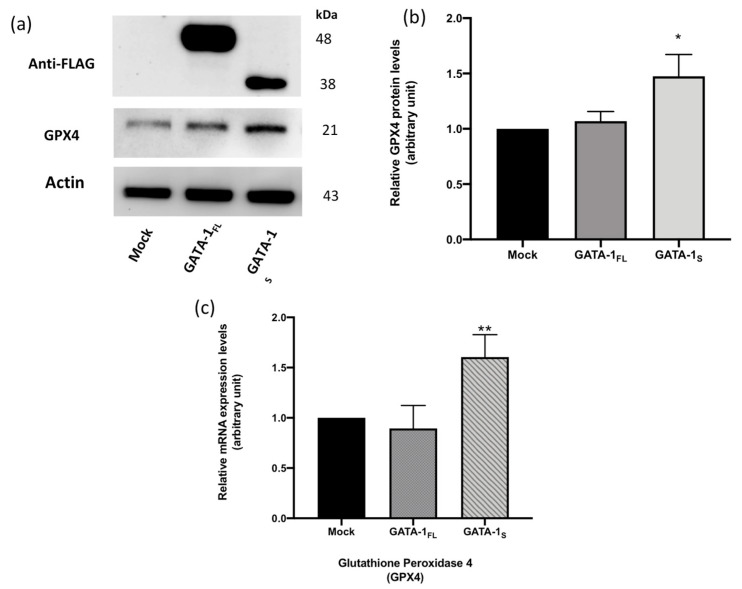
Overexpressed GATA-1_S_ is associated with enhanced GPX4 protein and mRNA levels. (**a**–**c**) Western blot analysis showing overexpressed FLAG-tagged GATA-1_FL_ and GATA-1_S_ isoforms and GPX4 expression levels in total protein extracts obtained from GATA-1_FL_ and GATA-1_S_ cells and from a mock control, respectively. Representative image of three independent experiments is shown. (**b**) Densitometric analysis of western blot results showing increased GPX4 levels only in K562 cells overexpressing the GATA-1_S_ isoform. For each sample, band intensities were quantified from three independent experiments, and normalized to actin used as a loading control. (**c**) Quantitative real-time PCR analysis of GPX4 mRNA in cells overexpressing GATA-1 isoforms and in a mock control. mRNA expression levels were normalized against actin. Results showed increased GPX4 mRNA levels in cells overexpressing GATA-1_S_, according to western blot analysis. All data represent the mean ± SD of three independent experiments. Statistical analysis was performed by one-way ANOVA, followed by Dunnett’s multiple comparisons test, where appropriate. * *p*-value ≤ 0.05, and ** *p*-value ≤ 0.001 versus mock control.

**Figure 5 antioxidants-12-00537-f005:**
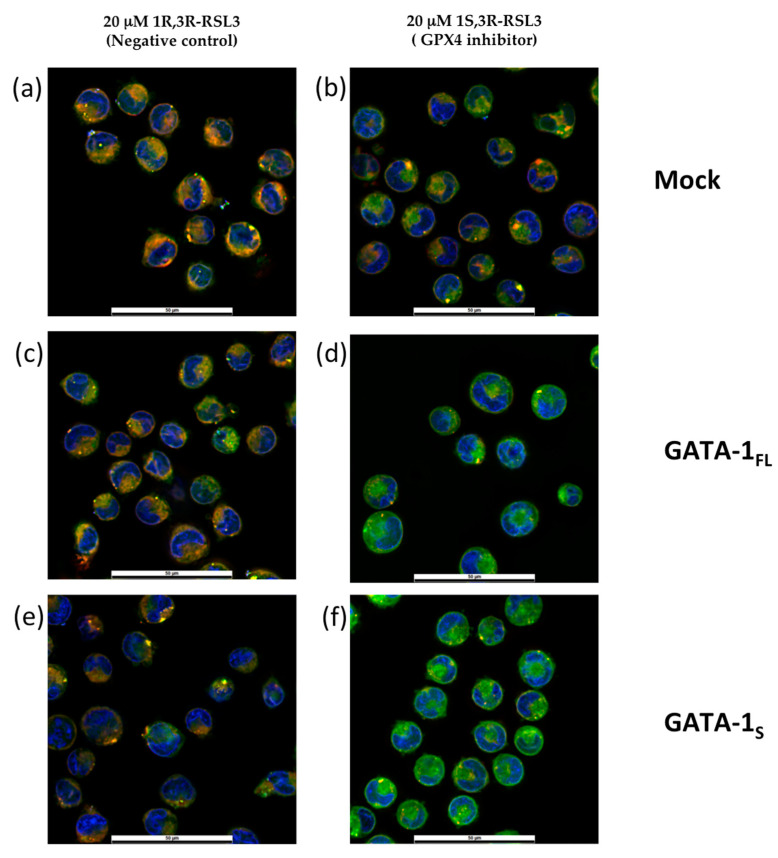
Effects of GPX4 inhibition on lipid peroxidation in K562 cells overexpressing GATA-1 isoforms. (**a**–**f**) Representative fluorescence merged images of K562 cells stained with BODIPY™ 581/591 C11 reagent for measurement of cellular lipid peroxidation after exposure to RSL3 for 24 h: the red to green shift corresponds to lipid peroxidation. Nuclei were stained with Hoechst 33342. (**a**,**c**,**e**) Cells treated with 20 μM 1R,3R-RSL3 (inactive enantiomer of 1S,3R-RSL3) were used as negative control. (**b**,**d**,**f**) Cells treated with 20 μM 1S,3R-RSL3 (glutathione peroxidase 4 inhibitor). (**g**) Quantitative fluorescence ratio analysis of green signals at 527 nm (representing peroxidized lipids)/red signals at 590 nm (representing nonperoxidized lipids) quantified by ImageJ as elsewhere reported [20,21]. Scale bar = 50 μm. All data shown are representative of three independent experiments. Statistical analysis was performed by one-way ANOVA, followed by Dunnett’s multiple comparisons test, where appropriate. * *p*-value ≤ 0.05, and ** *p*-value ≤ 0.001 compared to mock control; # *p*-value ≤ 0.05, GATA-1_FL_ cells versus GATA-1_S_ cells.

**Figure 6 antioxidants-12-00537-f006:**
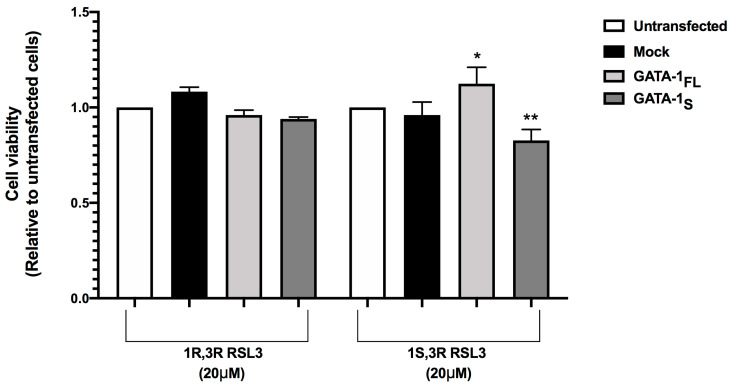
Effects of GPX4 inhibition on cell viability in GATA-1_FL_ and GATA-1_S_ cells. The effects of treatment with 20 μM 1R,3R RSL3 (negative control) and 20 μM 1S,3S RSL3 (GPX4 inhibitor) were evaluated with an MTT assay with respect to untransfected cells and mock control. In this context, it should be underlined that statistical significance of MTT signals in GATA-1_FL_ cells relative to untransfected and mock controls was of limited significance (* *p* value = 0.047), whereas a more robust significant difference was observed between GATA-1_S_ cells and untransfected and mock controls (** *p* value = 0.0046). The graph represents the mean and SD of three separate experiments.

**Figure 7 antioxidants-12-00537-f007:**
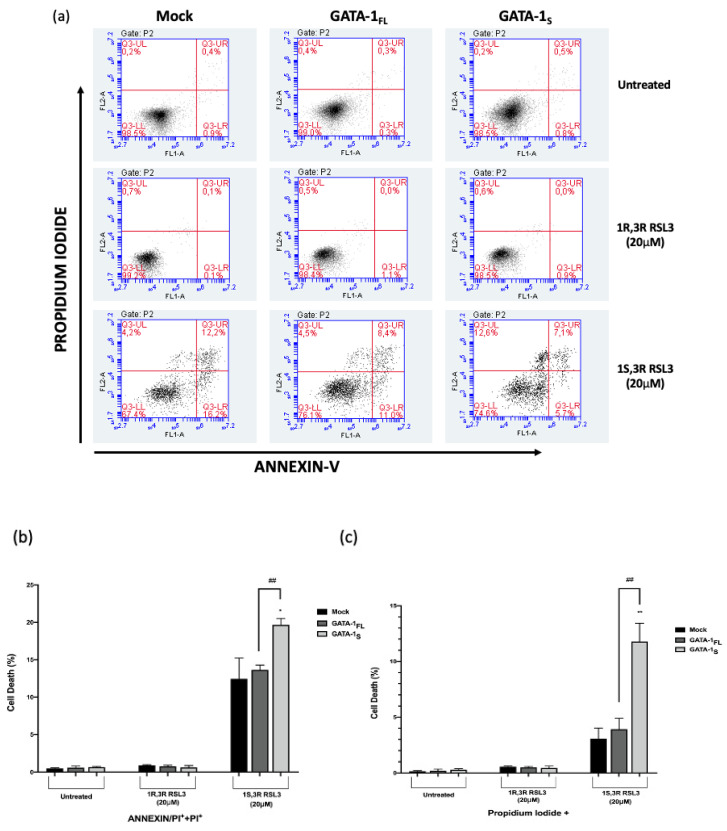
Effects of GPX4 inhibition on cell death in K562 cells overexpressing GATA-1 isoforms. (**a**) Representative images of flow cytometric evaluation of cell death. Necrotic and apoptotic cells were detected by annexin V and PI staining followed by flow cytometry analysis 48 h after transfection and 24 h of RSL3 treatment. The LR, UR, and UL quadrants show the annexin V+/PI− (early apoptosis), annexin V+/PI+ (late apoptosis/necrosis), and annexin V–/PI+ (necrosis), respectively. The LL quadrants measure the percentage of double-negative cells. No variation in annexin-V and PI percentage was observed in cells treated with the inactive enantiomer of RSL3 (1S,3R RSL3) compared to untreated cells. In contrast, the treatment with 20 μM 1S,3R RSL3 resulted in enhanced necrotic pathways in GATA-1_S_ cells; (**b**) Cumulative percentage of annexin-V+/PI+ and annexin-V–/PI+ cells; (**c**) Percentage of annexin-V–/PI+ cells. The graphs represent the mean and SD of three independent experiments. Statistical analysis was performed by one-way ANOVA, followed by Dunnett’s multiple comparison test, where appropriate. * *p* < 0.05, ** *p* < 0.0001 versus mock control, ## *p* < 0.0001, GATA-1_S_ cells versus GATA-1_FL_ cells.

**Figure 8 antioxidants-12-00537-f008:**
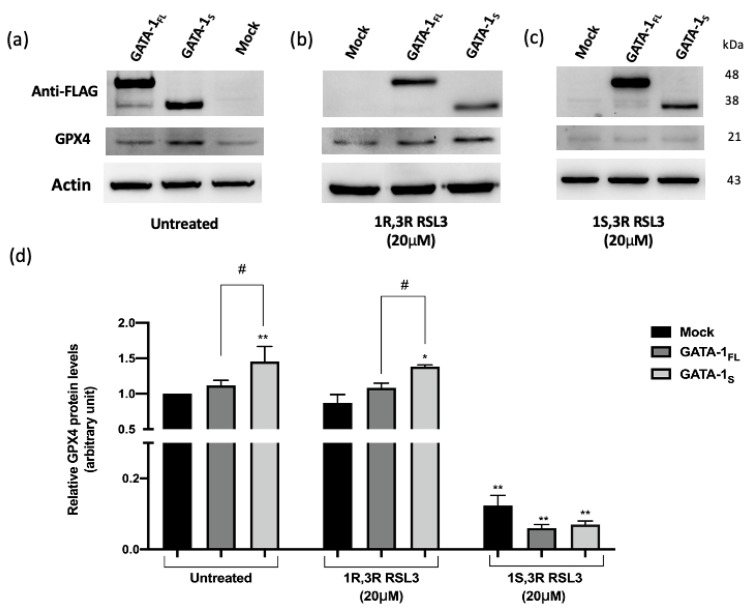
Effects of RSL3 treatment on GPX4 levels. (**a**–**c**) Representative western blot images and densitometric analysis of GPX4 levels in total protein extracts obtained from cells overexpressing FLAG-tagged GATA-1_FL_ or GATA-1_S_ and from mock control in (**a**) untreated cells and (**b**,**c**) cells treated with 1R,3R-RSL3 (inactive enantiomer, used as negative control) or with 1S,3R-RSL3 (glutathione peroxidase 4 inhibitor); (**d**) Densitometric quantification of western blot results showing higher GPX4 levels accompanied by GATA-1_S_ overexpression in untreated cells or cells treated with the RSL inactive enantiomer compared to GATA-1_FL_. Conversely, in all cell types (mock, GATA_1_FL_ cells and GATA_1_S_ cells) GPX4 levels were dramatically reduced after treatment with 1S,3R RSL3. For each sample, band intensities of the GPX4 signal were quantified and normalized to actin used as a loading control. All data represent the mean ± SD of three independent experiments. (**e**) Quantitative RT-PCR (qRT-PCR) analysis of GPX4 after treatment with RSL3. mRNA expression levels were normalized against β-actin. Results showed a more dramatic reduction of GPX4 mRNA levels in GATA-1_S_ cells treated with RSL3. Statistical analysis was performed by one-way ANOVA, followed by Dunnett’s multiple comparisons test, where appropriate. For each group * *p*-value ≤ 0.05, and ** *p*-value ≤ 0.001 compared to corresponding mock control, # *p*-value ≤ 0.05, GATA-1_FL_ cells versus GATA-1_S_ cells.

**Table 1 antioxidants-12-00537-t001:** Primer sequences used for quantitative real-time PCR analysis.

Transcript	Accession Number	Primer	Sequence 5′-3′	Amplicon Size
GPX4	NM_002085.5	For	CCTGGACAAGTACCGGGGC	140 bp
		Rev	CTTCGTTACTCCCTGGCTCCT	
β-actin	NM_001101.5	For	CGACAGGATGCAGAAGGAGA	160 bp
		Rev	CGTCATACTCCTGCTTGCTG	

## Data Availability

Not applicable.

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
