# Peer review of "Over-Expressed GATA-1S, the Short Isoform of the Hematopoietic Transcriptional Factor GATA-1, Inhibits Ferroptosis in K562 Myeloid Leukemia Cells by Preventing Lipid Peroxidation"

_antioxidants, 2023, doi:10.3390/antiox12030537_

Round 1
Reviewer 1 Report
Silvia Trombetti and co-authors in this manuscript explore the regulation of lipid perixidation-iduced ferroptosis with the final aim to propose ferroptosis as a strategy to overcome drug resistance in hematological malignancies. Ferroptosis is a cell death process due to the accumulation of lipid perixidation associated to the loss of function of antioxidant systems such as that mediated by GPX4, whose up-regulation is associated to ferroptosis’s suppression and to poor prognosis in myeloid leukemia. Targeting GPX4 may therefore increase ferroptosis. GATA-1 is a key transcriptional regulator of hematopoiesis-related genes and the over-expression of its shorter variant, GATA1s, is a poor prognostic factor in myeloid leukemia. Since authors previously showed that GATA1s is involved in K562 cell survival and apoptosis resistance, they now study its involvement in peroxidation-induced ferroptosis.
General comment
In some part the experimental rational and design and the related results are not very clear. Conclusion that GATA1s isoform overexpression is involved in response to ferroptosis is not sustained by the presented results. Furthermore the main aim of the study to propose ferroptosis as a strategy to overcome drug resistance is not explored.
Major concerns
1. I miss the clear connection between GATA1 and GPX1, is GPX1 one of the gene whose transcription is directly regulated by GATA1? If this is not the case the conclusions drown by the authors cannot be considered mechanistic.
2. Another important weakness of the manuscript is the use of a single cell line; finding obtained on the K562 only, cannot be generalized to hematological malignancies.
3. In Figure S1 is shown the relative expression of GATA1FL and GATA1s as detected by and FLAG antibody following their transfection in K562. Which is the basal expression level of these 2 isoforms in K562? The same concern can be applied to Figure 1, the IR spectrum profile of K562-mock transfected cells is missing.
4. Apparently the results reported in Figure S2 are referred to only 2 experiments (the 2 batches?) including 2 replicates of K562 transfected with Mock, GATA1FL and GATA1s. Considering the great variability observed between the 2 experiments, performing more analyses would probably have helped in results interpretation. The results adjustment performed with the tool ASCA is not very clear, nor the interpretation of the results. To have a good separation between the GATA1FL and GATA1s groups, the Mock group needed to be removed from the analysis.
5. Results shown in figures 3 and 5 are only qualitative and referred to only 1 representative experiment. A more quantitative way to represent the data should be used. The results are particularly questionable in figure 5, where the difference in panels d and f is not so evident. Could the Image-iT Lipid Peroxidation Sensor analyzed also by flow cytometry? If this is the case, the data should be shown.
6. In figure 6 the effect of 1S,3R RSL3 on GATA1FL apparently results in a cell growth advantage, therefore the differences between GATA1FL and GATA1s is questionable, a further control could be the analysis of cell viability performed also on untransfected K562, to allow a direct comparison of the 2 transfected cells with Mock transfection.
7. In the experiment reported in figures 7, 8 and 9 the effect of 1R,3R RSL3 (negative control) is missing and it must be shown.
8. It is unclear why authors measure apoptosis (with questionable results) instead of detecting variation in ferroptosis. Considering both early and late apoptosis detected in LR and UR quadrants, respectively in Figure 7a, the apoptotic effect induced by 1S,3R RSL3 was much greater in Mock transfected cells than in cells transfected with the GATA1 isoforms. This appears in contradiction with the data reported in panel b. Furthermore, since the effect of 1S,3R RSL3 on GPX4 protein expression in all transfected cells appears equivalent (Figure 8b), one would not expect differences in their apoptotic level following treatment with the GPX4 inhibitor.
9. It is unclear why the GPX4 inhibitor should affect also expression of GPX4 transcript.
Minor concern
1. Rational for performing infrared spectroscopy analysis should be made clear. Which should be the readout of such analysis?
2. What RI NIST in Table S1 stands for? Values of Delta RI are only intuitively derived from the difference between RI NIST and RI, this should be clearly stated and the meaning of Delta RI explained.
3. For a better interpretation of Figure S2 it should be better to associate the color code in panels A and B with the group they refer to. The same should be done for panel A in Figure S3 and in left panel of Figure 2.
Author Response
We thank the Reviewers for their careful reading of the manuscript and their constructive remarks. We have taken the comments on board to improve and clarify the manuscript. Please find below a detailed point-by-point response to all comments. Since the reordering and restructuring of the manuscript was substantial, we have written bullet points of our major changes to the manuscript, as shown in the enclosed ‘track changes’ document.
Reviewer 1
Silvia Trombetti and co-authors in this manuscript explore the regulation of lipid perixidation-induced ferroptosis with the final aim to propose ferroptosis as a strategy to overcome drug resistance in hematological malignancies. Ferroptosis is a cell death process due to the accumulation of lipid peroxidation associated to the loss of function of antioxidant systems such as that mediated by GPX4, whose up-regulation is associated to ferroptosis’s suppression and to poor prognosis in myeloid leukemia. Targeting GPX4 may therefore increase ferroptosis. GATA-1 is a key transcriptional regulator of hematopoiesis-related genes and the over-expression of its shorter variant, GATA1s, is a poor prognostic factor in myeloid leukemia. Since authors previously showed that GATA1s is involved in K562 cell survival and apoptosis resistance, they now study its involvement in peroxidation-induced ferroptosis.
General comment
In some part the experimental rational and design and the related results are not very clear. Conclusion that GATA1s isoform overexpression is involved in response to ferroptosis is not sustained by the presented results. Furthermore the main aim of the study to propose ferroptosis as a strategy to overcome drug resistance is not explored.
Response: We have made substantial changes in several part of the paper to address the Reviewer comments. We agree with the Reviewer comment that exploring the role of ferroptosis induction is a promising strategy to overcome drug resistance in leukemia, however this is beyond the scope of this study. Anyway, we are thankful to the Reviewer for this suggestion and we plan to explore this issue in more depth in the next future.
Major concerns
- I miss the clear connection between GATA1 and GPX1, is GPX1 one of the gene whose transcription is directly regulated by GATA1? If this is not the case the conclusions drown by the authors cannot be considered mechanistic.
Response: We are thankful to the reviewer for this pertinent comment. We agree that this is an important aspect that should be clarified. Motivated by the reviewer’s comment, we interrogated the GeneCards resource (www.genecards.org), an integrative approach used to retrieve detailed information of human genes. Annotated information from this database revealed no direct involvement of GATA-1 in GPX4 gene expression. However, we would put to your consideration that, in accordance with a large body of literature data (as an example: Zhou, et al. J Ovarian Res 2022; Byrska-Bishop et al, J Clin Invest. 2015; Rylski et al, Mol Cell Biol 2005), given the extensive GATA-1-regulated network of gene activation and repression, GPX4 up-regulation specifically associated with GATA-1S can be explained by a mechanism of GATA-1-mediated indirect transcriptional regulation. Therefore, we do not agree with the opinion that in this case “the conclusions drown by the authors cannot be considered mechanistic”. A sentence has been added in the revised manuscript to better clarify this point (Discussion Section).
2.1 Another important weakness of the manuscript is the use of a single cell line;
Response: We wish to point out that, although from a general point of view we agree that the use of more that one cell line would have been preferable, the rationale of using only K562 cells is related to the combination of several specific features of these cells that make them the most suitable experimental cell model to study the effects of altered expression of GATA-1 isoforms: their constitutive endogenous expression of both GATA-1 isoforms (as now shown in revised Fig. S1), their higher efficiency of transfection compared to other hematopoietic cells lines and their unique capacity to differentiate into progenitors of either the granulocyte, erythrocyte or monocyte series (all of these differentiation pathways being GATA-1-dependent processes).
2.2 …finding obtained on the K562 only, cannot be generalized to hematological malignancies.
Response: We took the reviewer point and modified the title and the text accordingly by referring our findings to K562 cells.
3.1 In Figure S1 is shown the relative expression of GATA1FL and GATA1s as detected by and FLAG antibody following their transfection in K562. Which is the basal expression level of these 2 isoforms in K562?
Response: Thank you for this comment. The constitutive endogenous levels of both GATA-1 isoforms are now shown in revised Fig. S1.
3.2 The same concern can be applied to Figure 1, the IR spectrum profile of K562-mock transfected cells is missing.
Response: We are thankful to the reviewer this comment. As stated in the revised paragraph 3.1, Result section, we now clarify that the IR spectra shown in Figure 1 a, b of the GATA transfected cells have to be considered as differential spectra with respect to the spectrum of the mock sample, since the reported absorbance values have been obtained by subtracting the mock absorbance contribution. It can be noticed that there is an evident difference in the integral absorbance of the GATA-1FL and GATA-1S transfected cells in the 3500- 2550 cm-1 region, where the absorption might be ascribed to the presence of stretching vibration bands of lipid compounds. This observation suggests the existence of a link between expression of specific GATA-1 isoforms and differences in cell lipid profiles.
4.1 Apparently the results reported in Figure S2 are referred to only 2 experiments (the 2 batches?) including 2 replicates of K562 transfected with Mock, GATA1FL and GATA1s. Considering the great variability observed between the 2 experiments, performing more analyses would probably have helped in results interpretation.
Response: The referee raised a good point, however the use of chemometrics tools such as PCA and ASCA allowed to safely clean the data and extrapolate the same trends in both batches (biological replicates). Moreover, the change of specific fatty acids level was in line with our starting hypothesis so we felt confident about using the collected data.
4.2 The results adjustment performed with the tool ASCA is not very clear, nor the interpretation of the results.
Response: The Results paragraph concerning ASCA has been now rephrased to make it clearer.
4.3 To have a good separation between the GATA1FL and GATA1s groups, the Mock group needed to be removed from the analysis.
Response: We agree with the referee that removing the Mock group from the analysis improves the separation between GATA1FL and GATA1s groups, indeed the final conclusions were actually drawn from the ASCA analysis performed only on GATA1FL and GATA1s groups (lines 309-335). We have now recomputed the PCA showed in Figure S2, only including GATA1FL and GATA1s groups and it is reported in new Figure S3.
- Results shown in figures 3 and 5 are only qualitative and referred to only 1 representative experiment. A more quantitative way to represent the data should be used. The results are particularly questionable in figure 5, where the difference in panels d and f is not so evident. Could the Image-iT Lipid Peroxidation Sensor analyzed also by flow cytometry? If this is the case, the data should be shown.
Response: We are grateful to the Reviewer for this useful comment that allows us to improve the quality of our data presentation. In the revised manuscript we added quantitative analysis of lipid peroxidation signals for experiments depicted in Figures 3 and 5.
- In figure 6 the effect of 1S,3R RSL3 on GATA1FL apparently results in a cell growth advantage, therefore the differences between GATA1FL and GATA1s is questionable, a further control could be the analysis of cell viability performed also on untransfected K562, to allow a direct comparison of the 2 transfected cells with Mock transfection.
Response: We agree with the reviewer’s comment that MTT results from RSL3 treatment could be interpreted as an apparent cell growth advantage to GATA1FL cells. Indeed, although several studies have revealed limitations of the MTT assay due to possible differences in cell redox capacities and/or mitochondrial metabolism that can lead to misinterpretations (as also reviewed by Ghasemi et al. The MTT Assay: Utility, Limitations, Pitfalls, and Interpretation in Bulk and Single-Cell Analysis. IJMS 2021, 22, 12827), it is commonly used to for representing treatment toxicity, cell viability, and metabolic activity. Therefore, on the basis of these observations, slight increased signal variations do not allow to definitely conclude on cell growth advantage. In this context, it is to be underlined that statistical significance in cell viability variation of GATA-1FL cells relative to untransfected and mock controls is of limited value (p value= 0.047) whereas a more robust significant difference has been observed between GATA-1 S cells relative to untransfected and mock controls (p value= 0.0046). With the aim to clarify this issue and avoid misinterpretations, we took the reviewer comment and added the required untransfected K562 cell control. The revised figure 6 includes this control and a new statistical analysis of these data. A sentence has also been added in the Results section to better clarify this point.
- In the experiment reported in figures 7, 8 and 9 the effect of 1R,3R RSL3 (negative control) is missing and it must be shown.
Response: We agree with the reviewer and modified the text and the figures by including the missed controls.
- It is unclear why authors measure apoptosis (with questionable results) instead of detecting variation in ferroptosis. Considering both early and late apoptosis detected in LR and UR quadrants, respectively in Figure 7a, the apoptotic effect induced by 1S,3R RSL3 was much greater in Mock transfected cells than in cells transfected with the GATA1 isoforms. This appears in contradiction with the data reported in panel b. Furthermore, since the effect of 1S,3R RSL3 on GPX4 protein expression in all transfected cells appears equivalent (Figure 8b), one would not expect differences in their apoptotic level following treatment with the GPX4 inhibitor.
Response: We apologize for the misleading data representation and are grateful to the reviewer for this comment that gives us the opportunity to clarify an important aspect of the mechanisms underlying ferroptotic-mediated cell death processes that had not been taken into account in the original version. With the aim to better clarify this point, we now specify that ferroptosis is a form of regulated necrotic cell death that can be identified through propidium iodide staining. Thus, instead of reporting the misleading cumulative percentages of annexin-V+ /PI+ and annexin-V- /PI+ cells to indicate the rate of necrotic cells, we now report annexin-V+ /PI+ and annexin-V- /PI+ cells on separate graphs. This graphical representation more clearly indicates the significant increase of necrotic cells in GATA-1S after RSL3 treatment which are largely to be ascribed to non-apoptotic cell death processes compared with mock and GATA-1FL cells (Fig. 7). Anyway, although our data indicating significant GPX4 down-regulation and lipid peroxidation, key hallmarks of ferroptosis, unequivocally support ferroptotic-regulated cell-death processes, it is well established that the induction of a given cell death pathway may promote the activation of other related cell death processes, including apoptosis and this can explain the relative enrichment in annexin-V +PI- cells, more evident in mock and GATA-1FL cells. A sentence in the Results section has been added to clarify this point.
- It is unclear why the GPX4 inhibitor should affect also expression of GPX4 transcript.
Response: We are thankful to the reviewer for giving us the opportunity to clarify this point. As reported in literature, inhibitory effects mediated by RSL3 comprise reduced activity along with protein and transcriptional levels of GPX4 (Bersuker et al. Nature 2019). Therefore, our findings are in line with these observations. The sentence regarding this issue in the Results section has been rephrased to better clarify this point.
Minor concerns
- Rational for performing infrared spectroscopy analysis should be made clear. Which should be the readout of such analysis?
Response: We are thankful to the reviewer for giving us the opportunity to clarify this point. As extensively reviewed by Bellisola and Sorio in 2012, according to the evidence that the transition from normal to cancerous phenotype is associated with biochemical, molecular and morphological changes, great efforts have focused toward the implementation of optical techniques capable of determining dimension, shape, or other cell physical properties for early detection of neoplastic changes so as to bring improvements in survival and clinical outcomes in cancer patients. Therefore, the rationale of our study is based on this evidence with the aim to identify novel oncometabolites and biomarkers linked to dysregulated GATA-1 expression. We have so far exploited the use of optical techniques in the 300 to 800 nm wavelength range. This approach has led us to highlight the different impact of GATA-1 isoforms on the production of mitochondrial and cytoplasmatic ROS as well as on the modulation of the expression levels of subunit C of the succinate dehydrogenase complex (SDHC). Given the role played by ROS in myeloid leukemogenesis, these observations also prompted us to evaluate the correlations between oxidative stress conditions triggered by GATA-1 isoforms and cell proliferation and apoptotic resistance (Riccio et al, J Cell Physiol, 2019). In the current study, we focused our attention on spectra differences in the IR region. As already discussed in the Results and Discussion sections, according to Bellisola and Sorio, variations in IR spectra prompted us to hypothesize different lipid profiles associated with the expression of specific GATA-1 isoforms and possibly a role in ferroptosis sensitivity in these cells. Lipidomic, cellular and molecular analyses have provided experimental evidence to our starting hypothesis. Accordingly, a sentence regarding this issue has been added in the revised Discussion section.
- What RI NIST in Table S1 stands for? Values of Delta RI are only intuitively derived from the difference between RI NIST and RI, this should be clearly stated and the meaning of Delta RI explained.
Response: As properly suggested by the reviewer, the description of the retention index (RI) and ΔRI has now been added in the Materials and Methods section.
- For a better interpretation of Figure S2 it should be better to associate the color code in panels A and B with the group they refer to. The same should be done for panel A in Figure S3 and in left panel of Figure 2.
Response: We absolutely agree with the reviewer, thus we have now clarified the color code in all the following figures: Figure 2, Figure S2, new Figure S3, new Figure S4 as suggested.
Reviewer 2 Report
Ferroptosis is a recently described cell death process caused by the accumulation of lipid hydroperoxides and inactivity of antioxidant systems like GPX4. In the myeloid leukemia context, upregulation of GPX4 correlates with poor prognosis. Therefore, targeting ferroptosis is emerging as a novel promising area of research in hematological malignancies. The authors indicate that over-expressed GATA-1S prevents myeloid leukemia cells from lipid peroxidation-induced ferroptosis and could eventually acts as a novel potential therapeutic target for hematological malignancies. Even if comprehensive and innovative some points need to be clarified and improved:
1) Infrared absorbance spectra results could be performed in presence of a control (e.g., empty k562 and k562 after treatment with GPX4 inhibitor), to set up which could be considered higher and lower levels and if these differences are significant. Moreover it could confirm the results obtained in figure 5.
2) Figure 3 needs a quantification of green and red signals to confirm the significance of the result.
3) Western blot in figure 4 must show in the same sample the band corresponding to transfected GATA-1 isoforms. Without this picture all the findings are not relevant.
4) Figure 5 needs as figure 3 the quantification. The only picture is not enough to draw any conclusion.
5) In figure 7 the scatter plot (A) didn’t correspond with the graph (B). Indeed, the percentage of GATA-1 FL and GATA-1S are reversed. So, which is the correct one?
6) Figure 8 and 9 could be merged with figure 7.
7) Finally, what about p53 status after transfection of GATA-1 isoforms? Since p53 is associated with ferroptosis, why did you only use K562 that are p53 mutated and not a cell line like MOLM-13, p53 wild type?
Author Response
We thank the Reviewers for their careful reading of the manuscript and their constructive remarks. We have taken the comments on board to improve and clarify the manuscript. Please find below a detailed point-by-point response to all comments. Since the reordering and restructuring of the manuscript was substantial, we have written bullet points of our major changes to the manuscript, as shown in the enclosed ‘track changes’ document.
Reviewer 2
Ferroptosis is a recently described cell death process caused by the accumulation of lipid hydroperoxides and inactivity of antioxidant systems like GPX4. In the myeloid leukemia context, upregulation of GPX4 correlates with poor prognosis. Therefore, targeting ferroptosis is emerging as a novel promising area of research in hematological malignancies. The authors indicate that over-expressed GATA-1S prevents myeloid leukemia cells from lipid peroxidation-induced ferroptosis and could eventually acts as a novel potential therapeutic target for hematological malignancies. Even if comprehensive and innovative some points need to be clarified and improved:
- Infrared absorbance spectra results could be performed in presence of a control (e.g., empty k562 and k562 after treatment with GPX4 inhibitor), to set up which could be considered higher and lower levels and if these differences are significant. Moreover it could confirm the results obtained in figure 5.
Response: We are grateful to the Reviewer for this useful comment. As stated in the revised paragraph 3.1, Result section, we now report that the IR spectra shown in Figure 1 a, b of the GATA transfected cells have to be considered as differential spectra with respect to the spectrum of the mock sample, since the reported absorbance values have been obtained by subtracting the mock absorbance contribution. It can be noticed that there is an evident difference in the integral absorbance of the GATA-1FL and GATA-1S transfected cells in the 3500- 2550 cm-1 region, where the absorption might be ascribed to the presence of stretching vibration bands of lipid compounds. This observation suggests the existence of a link between expression of specific GATA-1 isoforms and differences in cell lipid profiles.
Regarding IR analysis after treatment with GPX4 inhibitor, we are thankful to the reviewer for their insightful suggestions however, in this case, the sensitivity of the IR analysis would require a single-cell approach that is beyond the scope of this study.
- Figure 3 needs a quantification of green and red signals to confirm the significance of the result.
- Figure 5 needs as figure 3 the quantification. The only picture is not enough to draw any conclusion.
Response: We agree with this useful comment that allows us to improve the quality of our data presentation. In the revised manuscript we added quantitative analysis of lipid peroxidation signals for the experiments depicted in Figures 3 and 5.
- Western blot in figure 4 must show in the same sample the band corresponding to transfected GATA-1 isoforms. Without this picture all the findings are not relevant.
Response: We took the Reviewer comment and modified Figure 4 and Figure 8 accordingly.
- In figure 7 the scatter plot (A) didn’t correspond with the graph (B). Indeed, the percentage of GATA-1 FL and GATA-1S are reversed. So, which is the correct one?
Response: We apologize for the misleading data representation and are grateful to the reviewer for this comment that gives us the opportunity to clarify an important aspect of the mechanisms underlying ferroptotic-mediated cell death processes that had not been taken into account in the original version. With the aim to better clarify this point, we now specify that ferroptosis is a form of regulated necrotic cell death that can be identified through propidium iodide staining. Thus, instead of reporting the misleading cumulative percentages of annexin-V+ /PI+ and annexin-V- /PI+ cells to indicate the rate of necrotic cells, we now report annexin-V+ /PI+ and annexin-V- /PI+ cells on separate graphs. This graphical representation more clearly indicates the significant increase of necrotic cells in GATA-1S after RSL3 treatment compared with mock and GATA-1FL cells (Fig. 7).
- Figure 8 and 9 could be merged with figure 7.
Response: We partly took the reviewer comment. As additional panels have been included in revised Figure 7, to avoid overcrowded figures, we chose to merge only Figures 8 and 9 in revised Fig. 8 which now reports both protein and mRNA expression levels of GPX4 following RSL3 treatment.
- Finally, what about p53 status after transfection of GATA-1 isoforms? Since p53 is associated with ferroptosis, why did you only use K562 that are p53 mutated and not a cell line like MOLM-13, p53 wild type?
Response: We thank the reviewer for this pertinent comment and we agree that this is an important aspect that needs to be clarified. Indeed, since the discovery of this regulated cell death program, p53 has been identified as a relevant player in ferroptosis. However, mounting literature indicate p53-independent ferroptosis pathways in several experimental models. Therefore, our results obtained in p53-deficient K562 cells further supports the role of alternative pathways that may trigger or inhibit ferroptosis and the relevance of GATA-1 in regulating these processes. Motivated by the reviewer’s comment, we expanded the Discussion section and added references to better clarify this point. Furthermore, we would point out that MOLM-13 or several other leukemia cell lines do not constitutively express appreciable levels of GATA-1 proteins, thus limiting the significance of over-expression studies of GATA-1 isoforms. Our choice of K562 cells is thus related to the combination of several specific features including their constitutive endogenous expression of both GATA-1 isoforms (as now shown in revised Fig. S1), their higher efficiency of transfection compared to other hematopoietic cells lines and their unique capacity to differentiate into progenitors of either the granulocyte, erythrocyte or monocyte series (all of these differentiation pathways being GATA-1-dependent processes). For all the above reasons K562 cells represent the most suitable experimental model to study the effects of altered expression of GATA-1 isoforms. A sentence addressing this point has been added in the Conclusions section.
Reviewer 3 Report
The paper entitled: “Over-expressed GATA-1S, the short isoform of the hematopoietic transcriptional factor GATA-1, prevents myeloid leukemia cells from lipid peroxidation-induced ferroptosis” is a very well written that has great potential in terms of the direction of future research on hematological malignancies.
I have a few minor comments:
The abstract is a little too long. I would suggest shortening the introduction itself to one sentence related to the purpose of the research. Expand the issue of the methods used and the material for testing, which is very important in the case of hematological cancers. I would suggest in the method section describe the research model rather than prolong the introduction.
Research methods are described in detail. There are no comments for such an extensive description
The results are very well described and the discussion is concise and to the point.
Congratulations to the authors.
Author Response
We thank the Reviewers for their careful reading of the manuscript and their constructive remarks. We have taken the comments on board to improve and clarify the manuscript. Please find below a detailed point-by-point response to all comments. Since the reordering and restructuring of the manuscript was substantial, we have written bullet points of our major changes to the manuscript, as shown in the enclosed ‘track changes’ document.
Reviewer 3
The paper entitled: “Over-expressed GATA-1S, the short isoform of the hematopoietic transcriptional factor GATA-1, prevents myeloid leukemia cells from lipid peroxidation-induced ferroptosis” is a very well written that has great potential in terms of the direction of future research on hematological malignancies.
I have a few minor comments:
- The abstract is a little too long. I would suggest shortening the introduction itself to one sentence related to the purpose of the research. Expand the issue of the methods used and the material for testing, which is very important in the case of hematological cancers. I would suggest in the method section describe the research model rather than prolong the introduction.
Response: We amended the manuscript by shortening abstract and introduction Sections. According to the reviewer’s comment, more details regarding our experimental model were included in the Materials and Methods and Results sections.
- Research methods are described in detail. There are no comments for such an extensive description. The results are very well described and the discussion is concise and to the point. Congratulations to the authors.
Response: We are thankful to the reviewer and deeply appreciate their positive comments.
Round 2
Reviewer 1 Report
The authors extensively revised the manuscript however I am still concerned about important issues.
Authors state in the abstract: “Here, we provide the first evidence that over-expressed GATA-1S prevents K562 myeloid leukemia cells from lipid peroxidation-induced ferroptosis. Targeting ferroptosis is a promising strategy to overcome chemoresistance. Therefore, our results could provide novel potential therapeutic approaches and targets to overcome drug resistance in hematological malignancies.”, however they do not show any data confirming their hypothesis that targeting ferroptosis may overcome chemoresistance at least in K562 cell line.
To justify the use of a single cell line, authors specify that “K562 cells is […] the most suitable experimental cell model to study the effects of altered expression of GATA-1 isoforms: …..”. Considering K562 as a proof of principle, authors should show that their findings can be generalized to others models and are not just peculiar characteristics of this cell line.
Concerning the correlation between GATA1 and GPX1, the authors in discussion specify that “In this context, although there is no evidence that the GPX4 gene is a direct target of GATA-1 (GeneCards database, www.genecards.org), given the extensive GATA-1-regulated network of gene activation and repression, we speculate that GPX4 up-regulation associated with GATA1S can be explained by a mechanism of GATA-1-mediated indirect transcriptional regulation”. This is still a speculation not a mechanistic explanation, the indirect transcription regulation has not been demonstrated.
Author Response
Reviewer 1
The authors extensively revised the manuscript however I am still concerned about important issues.
1) Authors state in the abstract: “Here, we provide the first evidence that over-expressed GATA-1S prevents K562 myeloid leukemia cells from lipid peroxidation-induced ferroptosis. Targeting ferroptosis is a promising strategy to overcome chemoresistance. Therefore, our results could provide novel potential therapeutic approaches and targets to overcome drug resistance in hematological malignancies.”, however they do not show any data confirming their hypothesis that targeting ferroptosis may overcome chemoresistance at least in K562 cell line.
Response: We agree that exploring the role of ferroptosis in chemoresistance is of great relevance in cancer research. Indeed, this is emerging as a cutting-edge research topic as testified by the increasing number of reports in the last few years. Just to give an example, in the last month Wang et al reported a comprehensive review of existing literature on this topic (Overcoming cancer chemotherapy resistance by the induction of ferroptosis, Drug Resistance Updates, January 2023) and stated that “…Emerging studies have suggested that ferroptosis can regulate the therapeutic responses of tumours. Accumulating evidence supports ferroptosis as a potential target for chemotherapy resistance. Pharmacological induction of ferroptosis could reverse drug resistance in tumours……. Recent evidence suggests that ferroptosis is related to cancer chemotherapy resistance, and induction of ferroptosis can reverse drug resistance, which is a critical tumour suppression mechanism. Several oncogenic signalling pathways and tumour suppressors have been shown to suppress and promote ferroptosis. Gan et al. suggested that tumours have developed mechanisms to dictate susceptibility to ferroptosis, thereby ferroptosis could be therapeutically targetable in certain cancer types (Lei et al., 2022)….” Therefore, although this is still a working hypothesis that needs further investigations, it is clearly conceivable and reasonably acceptable that, in line with these lines of evidence, gaining and sharing new knowledge in this field “could provide novel potential therapeutic approaches and targets to overcome drug resistance in hematological malignancies” as we reported in the Abstract section. Anyway, according to these considerations, we aim to address this interesting topic in the next future.
2) To justify the use of a single cell line, authors specify that “K562 cells is […] the most suitable experimental cell model to study the effects of altered expression of GATA-1 isoforms: …..”. Considering K562 as a proof of principle, authors should show that their findings can be generalized to others models and are not just peculiar characteristics of this cell line.
Response: We would like to point out that we have already taken into account the previous reviewer comment regarding this issue (finding obtained on the K562 only, cannot be generalized to hematological malignancies) and we had modified the title and the text accordingly to fully address the reviewer request.
3) Concerning the correlation between GATA1 and GPX1, the authors in discussion specify that “In this context, although there is no evidence that the GPX4 gene is a direct target of GATA-1 (GeneCards database, www.genecards.org), given the extensive GATA-1-regulated network of gene activation and repression, we speculate that GPX4 up-regulation associated with GATA1S can be explained by a mechanism of GATA-1-mediated indirect transcriptional regulation”. This is still a speculation not a mechanistic explanation, the indirect transcription regulation has not been demonstrated.
Response: We agree that this is an important aspect that is worth further investigation. At the moment, based on the findings herein shown, we can only speculate on an indirect mechanism of regulation mediated by GATA-1 on GPX4. (regarding this point, we would respectfully specify that no correlation has been reported in our study between GPX1 and GATA-1) Anyway, as regards GPX4, our thought is in line with the well described complex network of GATA-1 target genes and with a large body of literature reporting several putative indirect targets of GATA-1 whose mechanisms of gene expression regulation still remain to be clarified (see, as an example, a recent paper by Trombetti et al, Antioxidants 2021, 10(10), 1603; https://doi.org/10.3390/antiox10101603 reporting abnormal expression of the subunit C of the succinate dehydrogenase complex associate with over-expression of GATA-1S). In any case, we would like to clarify that the demonstration of this mechanism of gene expression regulation is beyond the scope of the present study.

Reviewer 2 Report
The authors have satisfied my requests.
Author Response
Reviewer 2
Response: We are grateful to Reviewer 2 for their consideration and insightful comments which enabled us to improve our results and manuscript. We are really encouraged by the strengths they see in the manuscript and the contributions they believe our paper can make.
